# Improved Link Entropy with Dynamic Community Number Detection for Quantifying Significance of Edges in Complex Social Networks

**DOI:** 10.3390/e25020365

**Published:** 2023-02-16

**Authors:** Vasily Lubashevskiy, Seval Yurtcicek Ozaydin, Fatih Ozaydin

**Affiliations:** 1Institute for International Strategy, Tokyo International University, 1-13-1 Matoba-kita, Kawagoe 350-1197, Saitama, Japan; 2Department of Social and Human Sciences, Tokyo Institute of Technology, 2-12-1 Ookayama, Meguro-ku, Tokyo 152-8552, Japan; 3CERN, CH-1211 Geneva, Switzerland

**Keywords:** social networks, edge significance, link entropy, deep link entropy, Leiden, Louvain, Walktrap

## Abstract

Discovering communities in complex networks is essential in performing analyses, such as dynamics of political fragmentation and echo chambers in social networks. In this work, we study the problem of quantifying the significance of edges in a complex network, and propose a significantly improved version of the Link Entropy method. Using Louvain, Leiden and Walktrap methods, our proposal detects the number of communities in each iteration on discovering the communities. Running experiments on various benchmark networks, we show that our proposed method outperforms the Link Entropy method in quantifying edge significance. Considering also the computational complexities and possible defects, we conclude that Leiden or Louvain algorithms are the best choice for community number detection in quantifying edge significance. We also discuss designing a new algorithm for not only discovering the number of communities, but also computing the community membership uncertainties.

## 1. Introduction

Understanding which edges in a network are more significant than others from a specific perspective plays a key role not only in analyzing the network, but also in designing algorithms for manipulating it. An important perspective is the global connectivity and the spread of information. Whether a node is connected to or disconnected from the network can be easily figured out by computing the reachability matrix [1,2]. However, efficient flow among a network maintaining global connectivity is a problem beyond simple reachability, where its topology becomes crucial [3]. Quantifying the significance of nodes and edges has been studied in not only a general context [4,5,6], but also a variety of networks from power grids [7,8], communication [9] and wireless sensor networks [10,11] to biological networks [12,13] and co-citation networks in scientific publications [14,15]. Due to its importance, clustering of nodes towards forming communities [16,17] in biological networks has received particular attention, which is also a major issue in social networks [12,18], especially from a political perspective [19,20].

In online political communications, a social network is considered to consist of nodes clustered around political ideologies, or simply politically fragmented communities. A deliberative democracy cannot flourish where everyone thinks similarly, corresponding to a nonfragmented online society at one extreme, or at the other extreme with a highly fragmented society, resulting in no political polarization or a very sharp polarization, respectively [21,22]. Because an appropriate level of polarization is considered to be the driving force for reasonable democratic pluralism where a moderate clash of ideologies can take place. Group polarization is a natural consequence of politically fragmented communities [19], and depending on the dynamics of fragmentation and polarization, communities tend to form echo chambers [23,24] which might not weaken but further cement over time [25,26], potentially leading to extremism. Recently, Sasahara et al. have shown how unfriending accelerates the emergence of echo chambers [27]. The impact of every unfriending corresponding to removing the edges is not the same on the dynamics of polarization and echo chambers. Let us consider two edges, one connecting two nodes within the same community, and the other connecting nodes in two different communities. Intuitively, when compared to the nodes of the second edge, the unfriending of two nodes of the first edge has a limited impact on the propagation of information throughout the entire network.

According to Massumi’s affect theory, affect and perception are transitive among connected nodes instead of being confined within nodes [28]. Because some edges enable more efficient diffusion in a network than others, exploring critical or more significant edges is recognized as crucial in various disciplines.

For quantifying the edge significance in a complex network, following the betweenness centrality for nodes by Freeman [29] and for edges by Anthonisse [30] in the 1970s, based on the number of the shortest paths that go through an edge, Girvan and Newman proposed the edge betweenness centrality method [12]. Using the self-avoiding random walks of length *k*, Alahakoon et al. proposed the *k*-path centrality method [31] which was then generalized by De Meo et al. [32]. Wang et al. have proposed the degree product [33] and Cheng et al. have proposed the bridgeness [34] methods.

Information-theoretic and entropy-based functions are becoming more popular in network studies [35,36,37,38]. Bianconi defined and evaluated the structural entropy for studying the ensembles of networks [39], and Anand and Bianconi related Gibbs entropy to large deviations of conjugated canonical network ensembles [40]. Recently, degree distribution and degree remaining entropy were used to study network robustness [41,42]. Wang et al. studied rumor spreading models in networks using information entropy [43]. Wen et al. have used joint entropy to explore the vulnerability of transportation networks [44]. Based on Tsallis entropy, Lei and Cheong have proposed the Local Structure Entropy approach for ranking the significance of nodes [45].

In 2017, Qian et al. [46] proposed the Link Entropy (LE) method for quantifying edge significance on maintaining global connectivity. For an edge $e_{i-j}$ connecting nodes $n_{i}$ and $n_{j}$, following a community discovery step based on the iterative nonnegative matrix factorization (NMF) algorithm by Wang et al. [47] which computes the uncertainty of each node of belonging to each community, LE computes the entropies of $n_{i}$ and $n_{j}$, and their Jensen–Shannon divergence [48]. The basic idea is that edges which enable the interaction of communities are more significant. That is, the nodes $n_{i}$ and $n_{j}$ of such a community-bridging edge are connected to nodes belonging to more than one community. Therefore, when compared to nodes which are connected to nodes of only a single community, it is more uncertain to which community $n_{i}$ and $n_{j}$ belong to, resulting in a higher entropy.

Sorting edges in the descending order of their significance computed by each method listed above, and then removing the most significant edge one by one, Qian et al. calculates in each removal the fraction of nodes of the largest component as the KPI for comparing the performance of the methods. Faster decrease of the fraction from unity indicates a higher performance because it implies that the method has quantified the significance of edges more successfully regarding the global connectivity. This way, Qian et al. showed that their LE method outperforms the previously proposed methods [46].

Very recently, also taking into account the entropies of the adjacent nodes of $n_{i}$ and $n_{j}$ multiplied a weight χ to be determined, and running the community discovery algorithm at each removal, because the network keeps changing at each removal, the proposed Deep Link Entropy (DLE) method [49] was shown to outperform LE, or in the worst case with χ=0, it reduces to an improved version of LE by running the community discovery algorithm at each removal. It was also shown that DLE outperforms LE in detecting political secession [50]. Considering the recent work of Lv et al. on deep link prediction [51], this shows that deeper strategies have been becoming more popular in social network studies.

As will be detailed in the next section, in the above link entropy-based methods, the execution of the community discovery algorithm in each quantification step is initiated with a prefixed number of communities. This number is obtained from either a visual observation or a general knowledge on the initial topology of the given network. However, considering an arbitrary network in general, or even a well-known network, throughout the changes in the topology due to edge-removal, the pre-fixed number of communities might fail to grasp the actual number of communities. For example, a network might initially consist of four communities, but during the edge-removal process, or in real-life social networks, due to unfriending or unfollowing each other, some communities disintegrate, resulting in more than four communities.

In this work, we ask whether predicting the number of communities before running the community discovery algorithm each time can play an important role in determining the performance of the edge significance quantification methods. Employing three major community number detection algorithms, namely Louvain [52], Leiden [53] and Walktrap [54], we propose an improvement over the LE method, which we call the Improved Link Entropy (ILE) method. Running experiments on five popular benchmark networks, that is, Wang et al.’s network [47], Zachary’s Karate Club [55], Dolphins [56], Hermaphrodite [57], and Jazz [58], networks, we show that while running the community discovery algorithm in each edge-removal iteration as the basic ILE method is already a significant improvement over LE, detecting community numbers as an extra initial step can further significantly improve the performance of ILE.

This paper is organized as follows. In the Materials and Methods section, we first present the LE method consisting of the NMF algorithm of Ding et al. [47] and quantification strategy of Qian et al. [46]. Next, we briefly present the community number detection algorithms, our proposed ILE method, and how to test the performance of each method. In the Results section, we present our experimental findings showing the significant improvement of ILE. Following the discussions including prospects on online political communications, we conclude.

## 2. Materials and Methods

### 2.1. Nonnegative Matrix Factorization Algorithm

Let us consider an undirected unweighted network with *k* unconnected communities, and graph of each community is represented with adjacency matrix $S_{i}$. Then the adjacency matrix of the network can be represented by *G* as
(1)$$G=\left[\begin{array}{cccc} S_{1} & 0 & \cdots & 0\\ 0 & S_{2} & \cdots & 0\\ \vdots & \vdots & \ddots & \vdots \\ 0 & 0 & \cdots & S_{k} \end{array}\right],$$
which can be factorized as $G=XSX^{⊤}$ with k×k identity matrix *S* and the community membership matrix *X*.
(2)$$X=\left[\begin{array}{cccc} 1 & 0 & \cdots & 0\\ \vdots & 0 & \cdots & 0\\ 1 & 0 & \cdots & 0\\ 0 & 1 & \cdots & 0\\ \vdots & \vdots & \ddots & \vdots \\ 0 & 0 & \cdots & 1 \end{array}\right].$$

This simple factorization does not apply to the case of real networks where communities can be connected, so that *G* would not be diagonal, and columns of *X* would not be orthogonal. The nonnegative matrix factorization (NMF) algorithm of Wang et al. can be used for discovering communities [47], which was also used as the first step of the LE method [46]. Because the network is undirected, *X* can absorb *S* as $X\leftarrow XS^{1/2}$. Aiming to find *X*, NMF is based on minimizing the loss
(3)$$\min_{X\geq 0}{\left\|G-XX^{⊤}\right\|}_{F}^{2},$$
with the loss function
(4)$${\left\|A-B\right\|}_{F}^{2}=\sum_{ij}{\left(A_{ij}-B_{ij}\right)}^{2}.$$
Starting with a random *X* matrix of nonzero elements, the minimization can be performed by iteration
(5)$$X_{ik}\leftarrow X_{ik}\left(\frac{1}{2}+\frac{{\left(GX\right)}_{ik}}{{\left(2XX^{⊤}X\right)}_{ik}}\right).$$
and normalizing the rows of *X* in each iteration. After sufficient number of iterations, each $X_{ik}$ represents the probability that node *i* is a member of community *k*.

### 2.2. Edge Significance Quantification of Link Entropy Method

The above NMF algorithm finding the *X* matrix, that is, the probability of each node to be the member of each community constitutes the first step of the LE method proposed by Qian et al. [46]. To calculate the significance of an edge $e_{i-j}$ connecting nodes *i* and *j*, LE first takes the average $M=(X_{i}+X_{j})/2$.

With the standard entropy
(6)$$H\left(X_{i}\right)=-\sum_{k=1}^{K}x_{ik}\log x_{ik},$$
and
(7)$$D\left(X_{i}\|M\right)=\sum_{k=1}^{K}x_{ik}\log\frac{x_{ik}}{m_{k}},$$
LE calculates the Jensen–Shannong divergence
(8)$$JSD\left(X_{i}\|X_{j}\right)=\frac{D\left(X_{i}\|M\right)+D\left(X_{j}\|M\right)}{2}.$$
and finally, the significance of edge $e_{i-j}$ as
(9)$$LE_{ij}=\frac{\left(H(X_{i})\right)+\left(H(X_{j})\right)/2+JSD\left(X_{i}\|X_{j}\right)}{2}.$$

### 2.3. Testing the Performance of Edge Significance Methods

Following the LE method [46], to compare the performance of edge significance quantification methods, we use the fraction of nodes of the largest component, $R_{GC}$ as the KPI. Given that every node in the network is reachable, initially, $R_{GC}=1$. Having quantified the significance of all the edges according to each method, edges of the network were removed in a descending order of significance, and at each removal, $R_{GC}$ was re-calculated, which became $R_{GC}<1$ in the first disconnection of any node or community. $R_{GC}$ keeps decreasing as more nodes or communities are disconnected. With a network of *n* nodes, the disconnection of a single node yields $R_{GC}\to R_{GC}-1/n$, while the disconnection of a community of *m* nodes yields $R_{GC}\to R_{GC}-m/n$.

In some cases, plotting the curves showing the decrease of $R_{GC}$ according to each method provides a clear visual opportunity to compare the performance of each method. However, to have a numerical comparison, we calculate the area under each curve. Because approaching zero in the curve faster indicates better performance, the method which results in a smaller area outperforms others.

The significance of edges is defined from the perspective of global connectivity and propagation of information among the network. Therefore, edges constituting a bridge between communities yield greater significance, and their removal results in a greater decrease of $R_{GC}$. Therefore, the decrease pattern indicates whether a single node, or a community of a particular number of nodes is disconnected at each edge-removal. Moreover, s faster decrease of $R_{GC}$ approaching zero through quantification of a method shows that it outperforms other methods. A key point here is that LE [46] runs the community discovery algorithm only once at the beginning, and removes the edges one by one. However, removal of each edge changes the network. Considering this, besides taking into account the *deeper* nodes as well, DLE [49] runs the community discovery algorithm not only at the beginning, but also after each removed edge, as an improvement over the original LE.

### 2.4. Community Number Detection Algorithms

An important shortcoming of both LE and DLE is that to start the community discovery algorithm, they set the number of communities manually at the beginning based on a priori knowledge. Thus, they cannot grasp the actual number of communities throughout the edge-removal process. The network might start with *k* communities, but due to removing some edges, number of communities might be greater than *k*, degrading the performance of community discovery algorithm that finds *X*, the community membership matrix, and therefore the performance of edge significance quantification algorithms.

In this work, aiming to address this shortcoming, our method integrates the following community number detection algorithms to the community discovery algorithm.

#### 2.4.1. Louvain

Newman and Girvan proposed the modularity method to exploring the community structure in networks, which is based on maximizing the difference between the actual number of edges in a community and the expected number of such edges [59]. Although this method is very successful, Brandes et al. showed that its optimization is an NP-hard problem [60], motivating researchers to find efficient optimization algorithms such as hierarchical agglomeration by Clauset et al. [61], extremal optimisation by Duch and Arenas [62], and Louvain by Blondel et al. [52] being one of the best-performing and fastest algorithms.

The Louvain algorithm optimizes modularity as the quality function by repeating two phases until the quality function achieves its highest value. Starting with a single partition considering each node as a community, the first phase of the Louvain algorithm is the local moving of nodes, where individual nodes are moved to one of the communities which increases the quality function the most, resulting in a partition. The second phase is the aggregation, where the partition is used to create an aggregate network, and the nodes in the partitions are treated as nodes in the aggregate network.

#### 2.4.2. Leiden

Pointing out a major defect of the Louvain algorithm which may result in badly connected, and even disconnected communities, Traag et al. proposed the Leiden algorithm [53], fixing the defect and also running faster. The Leiden algorithm employs a refinement phase between the two phases of the Louvain algorithm. The last phase of the Leiden algorithm is based on aggregating the network based on refined partition, and the non-refined partition is used to create an initial partition of the aggregate network. The refinement was performed as follows. Unlike the Louvain algorithm which moves the nodes to a community resulting in the maximum increase of the quality function, the Leiden algorithm randomly selects the community to merge the node among the ones that yield an increase of the quality function. The probability of selecting each community is proportional to the expected increase in the quality function.

#### 2.4.3. Walktrap

To reduce the running time of exploring the community structure in complex networks, following the intuition that random walks on a graph frequently becomes stuck or “trapped” into sub-graphs with many connections corresponding to communities, Pons and Pataly proposed a measure of the similarity between nodes based on random walks [54]. The distances between nodes are computed through the random walks, and nodes are assigned to communities based on these distances. via bottom-up hierarchical clustering.

## 3. Results

For each benchmark network, to test the performance of our approach, we run five simulations as follows:

**LE:** Original Link Entropy method of Qian et al. [46].

**ILE:** Improved Link Entropy method which runs the community discovery algorithm in each iteration with a fixed number of community number.

**ILE_Louvain:** ILE which runs the community discovery algorithm after detecting the number of communities by the Louvain method in each iteration.

**ILE_Leiden:** ILE which runs the community discovery algorithm after detecting the number of communities by the Leiden method in each iteration.

**ILE_Walktrap:** ILE which runs the community discovery algorithm after detecting the number of communities by the Walktrap method in each iteration.

We refer to the last four methods together as ILEs.

In each iteration, we remove the most significant edge according to each method and calculate $R_{GC}$ as explained in Section 2.3. The method in which $R_{GC}$ approaches zero faster is more successful in quantifying the significance of edges. The amount of decrease of $R_{GC}$ indicates the number of nodes disintegrated in each iteration. A small decrease the disintegration of a single node or a few nodes, while a huge decrease indicates the disintegration of a community with many nodes.

The performances of three community discovery algorithms that we use to detect the number of communities, that is, Louvain, Leiden and Walktrap are usually compared with respect to their discovery capabilities, shortcomings or defects, and running times. Our work provides a new test bed for the performances of these algorithms in a real-life scenario of quantifying the edge significance for global connectivity and spread of information.

### 3.1. Wang et al.’s Network

Our first test was on the network considered by Wang et al. in their community discovery work (Figure 1 of the Ref. [47]). Consisting of 30 nodes and 75 edges with a maximum degree of 7, this is a smaller and less complex network when compared to the other networks used for testing the algorithms. In this network, LE starts disintegrating nodes or small communities much earlier than ILEs. However, ILEs start disintegrating large communities, resulting in huge sudden drops in the curves, as can be seen in Figure 1. Except for a few iterations in the beginning, all four ILEs outperform LE significantly, as can be seen by comparing the areas under each curve presented in Table 1. It is interesting that the performance of all ILEs almost perfectly overlaps throughout the process, implying that taking a constant number of communities (i.e., ILE) or detecting the number through three different algorithms in each iteration has no significant effect on the performance of quantifying edge significance in a relatively small network. Therefore, we continue testing our approach in larger and more complex networks.

### 3.2. Zachary’s Karate Club Network

Next, we tested our methods on the Zachary’s Karate Club network [55] which is very popular in testing various network analysis methods. The Karate Club network consists of 34 nodes and 78 edges, and is usually considered to have two communities with a maximum degree of 17. In this network, the sudden and huge drops of its curve show that LE can disintegrate communities as well. The performances of ILE (without detecting the number of communities), ILE_Louvain and ILE_Leiden almost perfectly overlap (see Figure 2, while ILE_Walktrap performs the worst in the beginning but catches other ILEs after removing around 20 edges. This result shows that as the network grows and becomes more complex (when compared to Wang et al.’s network, for example), ILEs start performing differently, increasing the motivation to perform experiments on more complex networks, which can be seen by comparing the areas under each curve presented in Table 2.

### 3.3. Dolphins Network

We now continue with a relatively larger and more complex network, that is, the Dolphins network [56], which consists of 62 nodes and 159 edges, and the maximum degree is 12. In the Dolphins network, LE immediately starts and keeps disintegrating single or a few nodes in each iteration, while the ILEs can initiate disintegration after a while, but disintegrates communities, as can be seen from the sudden large drops in the curves in Figure 3. Though the performance of ILEs approximately matches after removing 80 edges, the first community disintegration is achieved by ILE_Leiden, followed by ILE_Louvain, ILE_Walktrap, and finally by ILE (without community number detection). Besides the fact that even ILE outperforms LE by discovering communities in each iteration, because ILEs disintegrate not a single or a few nodes, but communities, it turns out that they outperform LE, ILE_Leiden achieving the best performance. Numerical results are presented in Table 3.

### 3.4. Hermaphrodite Network

Next, we ran our experiments on the Hermaphrodite Gap Junction Corrected network [57], which we call the Hermaphrodite network for short. This network consists of 469 nodes and 1450 edges. As shown in Figure 4, while ILEs start with a small decrease, LE starts with a huge decrease in the beginning. Following a constant $R_{GC}$ until the removal of around the 180th edge, similar to the case of the Dolphins network, LE starts to disintegrate one or a few nodes in each iteration, while ILEs can disintegrate the next few communities later than LE but start to outperform it significantly. In this network, ILE_Louvain performs the best, followed by ILE_Leiden, and ILE_Walktrap and ILE which achieve similar performances, as can be seen from Table 4. An important and interesting point about this result is that the refinement phase of the Leiden algorithm as a general improvement over Louvain yields worse performance in quantifying the edge significance problem in this network.

### 3.5. Jazz Network

Our last experiment is on the Jazz network [58] consisting of 198 nodes and 2742 edges with the maximum degree being 100. We found results similar to those of the Dolphins network, though ILE_Louvain and ILE_Leiden achieved almost the same performance with overlapping curves in Figure 5. The overlapping of ILE_Louvain and ILE_Leiden showed that the refinement phase of the Leiden algorithm as an improvement over the Louvain algorithm was ineffective in this network. Numerical results for the areas under the $R_{GC}$ curves are presented in Table 5.

In order to provide a comprehensive picture on the overall results, in Figure 6, we plotted the area under the $R_{GC}$ curve for each network found by ILEs normalized to LE. This shows that although the impact of community discovery algorithms on edge significance quantification is similar for small networks, ILE_Louvain and ILE_Leiden outperform Walktrap as the network grows. Considering the area under the $R_{GC}$ curve as the KPI, our ILE algorithm roughly doubled the performance of LE in general, especially if ILE_Louvain or ILE_Leiden were used.

The major contribution of our work is the improvement of the LE algorithm with dynamic community number detection, community discovery, and community membership probability updates in each edge-removal iteration. For this purpose, any community number detection algorithm can be used. Here, we chose three popular algorithms. Although all gave similar results in small networks, the difference became bigger for larger networks, suggesting the use of ILE_Louvain or ILE_Leiden. Our study does not exclude the possibility of achieving even better results by using another community number detection algorithm.

In the Ref. [46], Qian et al. found that regardless of the real number of communities, *k*, setting k=2 in the LE algorithm yields the best results. However, we show that discovering communities not only once in the beginning as in LE but in every iteration as in ILE, the value of *k* plays a crucial role in quantifying edge significance in complex networks.

## 4. Discussions

The reason why we used two types of community discovery algorithms in our method, that is, the set of three algorithms, namely Louvain, Leiden, and Walktrap, and the algorithm of Wang et al. [47] is as follows. First of all, in order to consider an entropy-based algorithm, the probability of each node to be the member of each community is required, which was provided by the algorithm of Wang et al. Secondly, in order to make a reliable comparison between the performance of each method and the performance of LE, the same community discovery algorithm, that is, the algorithm of Wang et al. must be used as the first stage. However, the shortcoming of that algorithm is that it requires the number of communities. Both LE and DLE uses the algorithm of Wang et al. with a pre-set number of communities, which is considered in this work as a shortcoming of those methods. This is because even if the initial number of communities in the given network is known, the topology and the adjacency matrix changes by creating new edges and removing some of the existing ones in general. In social networks, these actions correspond to friending or following, and unfriending or unfollowing, respectively. Specific to the performance testing of the edge significance quantification methods where each edge is removed in the descending order of significance quantified by each method, the topology keeps changing at each removal, potentially changing the community structure and the number of communities. Therefore, providing the actual number of communities to the community discovery algorithm of Wang et al. might have an impact on the performance of the later stage of quantifying the edge significance in any entropy-based method. In order to explore this impact, we propose to use the three community discovery algorithms to find the number of communities. Running experiments on popular benchmark networks in the ascending order of complexity, we show that the larger and more complex the network is, detecting the number of communities in each iteration yields better performance.

Although the tests of Traag et al. showed that the Leiden algorithm significantly outperforms Louvain in general thanks to the refinement phase [53], our results on the edge significance quantification problem show that it is not the case every time; that is, the refinement yields zero or a very small performance difference in the small Wang et al. network and Zachary’s Karate Club networks, and slightly worse performance then Louvain in the Hermaphrodite network.

Considering the two arguments above in addition to reducing the overall running time, we point out to the need to design new community discovery algorithms which not only discovers the communities such as Louvain, Leiden or Walktrap, but also calculates the community membership probability of each node, such as the algorithm of Wang et al., which does not require the number of communities as a priori knowledge.

The DLE method [49] was also proposed as an improvement over LE by taking into account, with a weight of χ, the uncertainties of the adjacent nodes of the pair of nodes connected by the edge in consideration. It also included the community discovery algorithm of Wang et al. The reason we have not studied DLE in this work is that although the weight was set to χ=−0.12 in the Ref. [49] which was sufficient to show that DLE outperforms LE, in general, the χ to be determined is specific to each social network. Therefore, we leave this problem to a future work.

We analyzed popular benchmark networks in the literature in a systematic way. Technically, our algorithm can be used for even larger networks. However, when it comes to social networks such as Facebook, consisting of billions of nodes and edges with a maximum degree up to 5000 [24], software and even hardware optimization can be considered for reducing the execution time.

The drawback of introducing additional steps to the entropic algorithm for significantly more precise edge significance quantification is the increase in the computational complexity. In the Ref. [54], Walktrap algorithm’s complexity was found to be $O(n^{2})$. Moving the nodes not to the best neighbor community as in the original Louvain algorithm, but to a random neighbor community, Traag reduced the computational complexity of the Louvain algorithm from $O(n^{2})$ to O(nlog(n)) [63]. The Leiden algorithm not only fixes some defects of the Louvain algorithm, but also runs faster [53]. Hence, considering also that ILE_Louvain and ILE_Leiden outperform ILE_Walktrap in edge significance quantification as shown in Figure 6, they appear as a better choice.

Nevertheless, designing a new algorithm combining NMF [47] and Louvain or Leiden in the sense that it discovers the number of communities and at the same time computes entropies based on community membership probabilities, we believe the overall computational complexity of ILE can be reduced significantly.

Thanks to the asymmetric version of the NMF algorithm of Wang et al. [47], in principle, both LE and our ILE algorithms can be adapted to directed graphs as well. However, because not only the diffusion mechanism but also the dynamics and interpretation of fragmentation and echo chambers become more sophisticated in directed graphs, we consider this problem for future works.

## 5. Conclusions

We have proposed a significant improvement over the Link Entropy (LE) method which has been outperforming the previous edge significance quantification algorithms. The LE method starts with the community discovery algorithm as the first phase, which requires a pre-set number of communities. Our method employs the Louvain, Leiden and Walktrap algorithms to detect the number of communities as the initial step, so that the community discovery algorithm yields more accurate results. Running experiments on popular networks in a spectrum of complexity, we have shown that the our proposal outperforms the LE method. We also showed that, although the Leiden algorithm was introduced to outperform the Louvain algorithm by employing a refinement phase, when it comes to an edge significance quantification task, in some networks, it performs similarly to the Louvain algorithm. We also pointed to the necessity of designing a new algorithm for discovering the community number detection and computing the community membership probabilities at the same time.

## Figures and Tables

**Figure 1 entropy-25-00365-f001:**
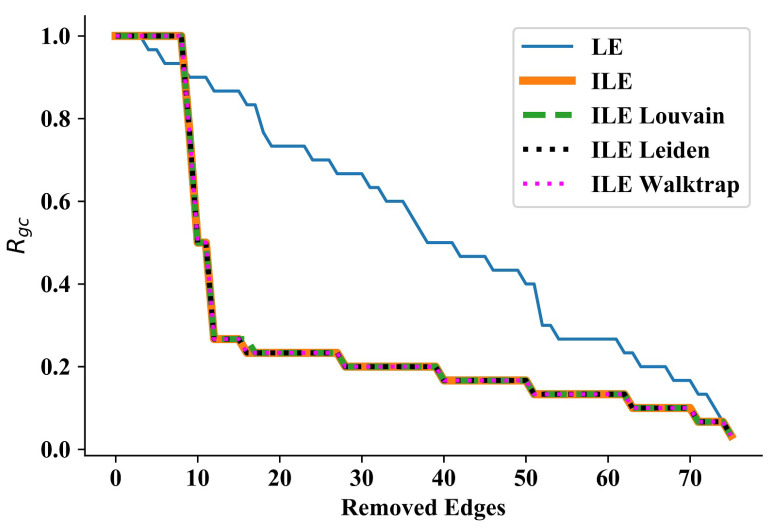
Results for Wang et al.’s network [47]. Detecting the number of communities via three algorithms in each iteration or not, ILEs perform almost the same, and due to discovering communities in each iteration, ILEs significantly outperform LE.

**Figure 2 entropy-25-00365-f002:**
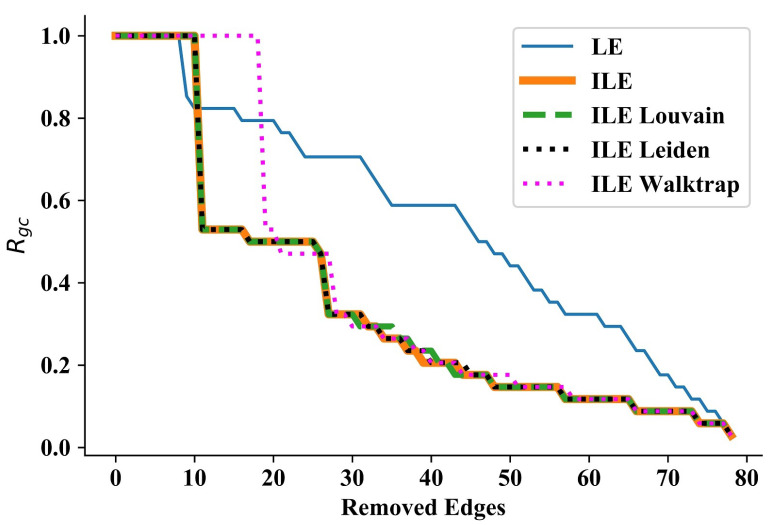
Results for the Zachary’s Karate Club network [55]. Thanks to discovering communities in each iteration, ILEs significantly outperform LE. Using a pre-set number or detecting the number in each iteration via Louvain or Leiden algorithms achieves almost the same performance throughout the process. Using the Walktrap algorithm starts disintegrating communities last, but catches the performance of other ILEs and keeps overlapping with them.

**Figure 3 entropy-25-00365-f003:**
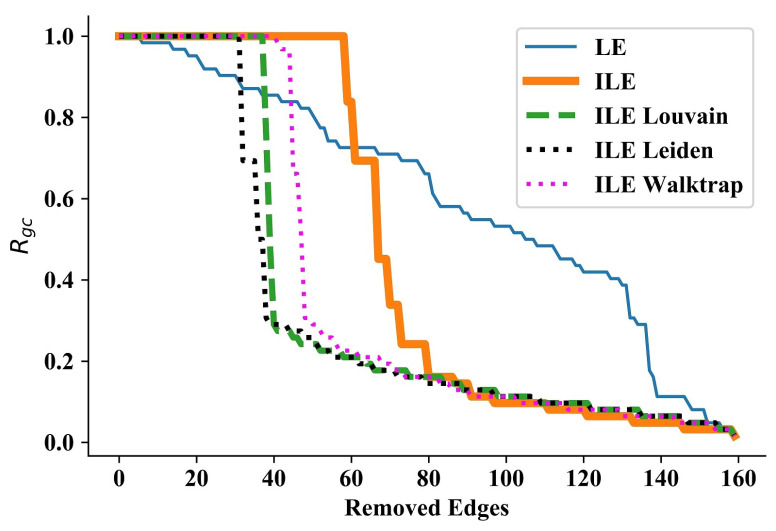
Results for Dolphins network [56]. The performances of ILEs start to differ as the network becomes more complex. In addition to outperforming LE, the best being the ILE_Leiden, all ILEs disintegrate large communities.

**Figure 4 entropy-25-00365-f004:**
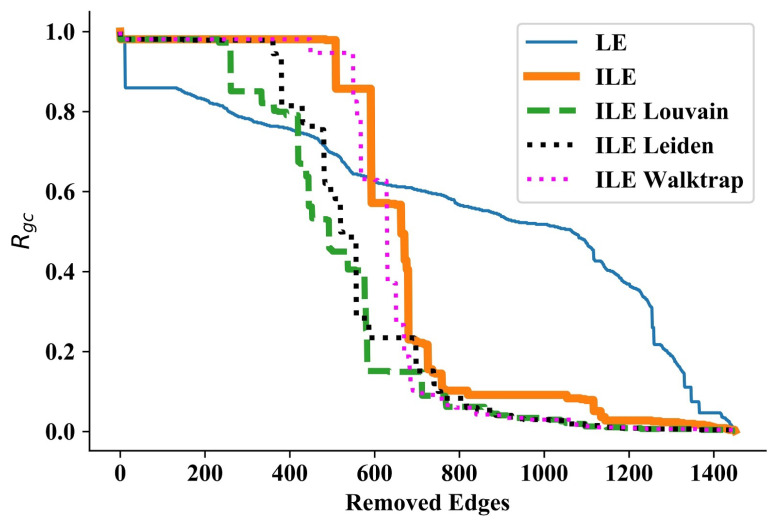
Results for the Hermaphrodite network [57]. Although LE starts better by disintegrating a larger community than ILEs in the beginning, it is outperformed by ILEs afterwards.

**Figure 5 entropy-25-00365-f005:**
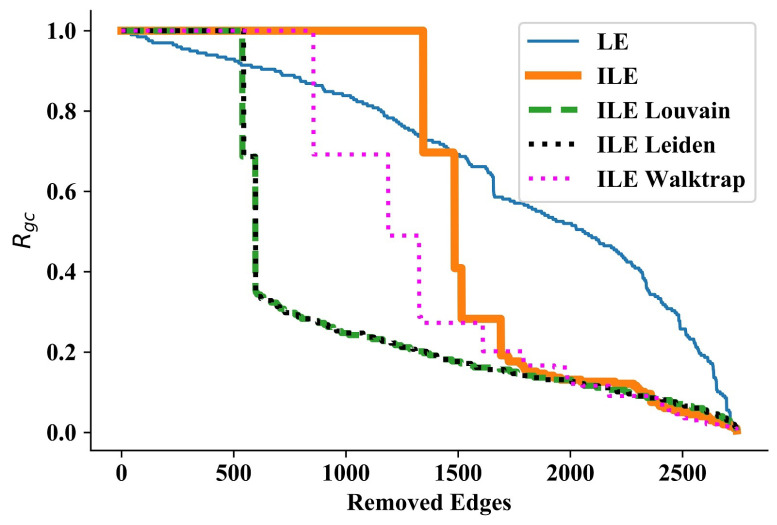
Results for the Jazz network [58]. Except for a few times, the LE method can disintegrate only one node or a few nodes, but not large communities consisting of many nodes, and it is outperformed by ILEs, the best by detecting the numbers via the Louvain and Leiden algorithms.

**Figure 6 entropy-25-00365-f006:**
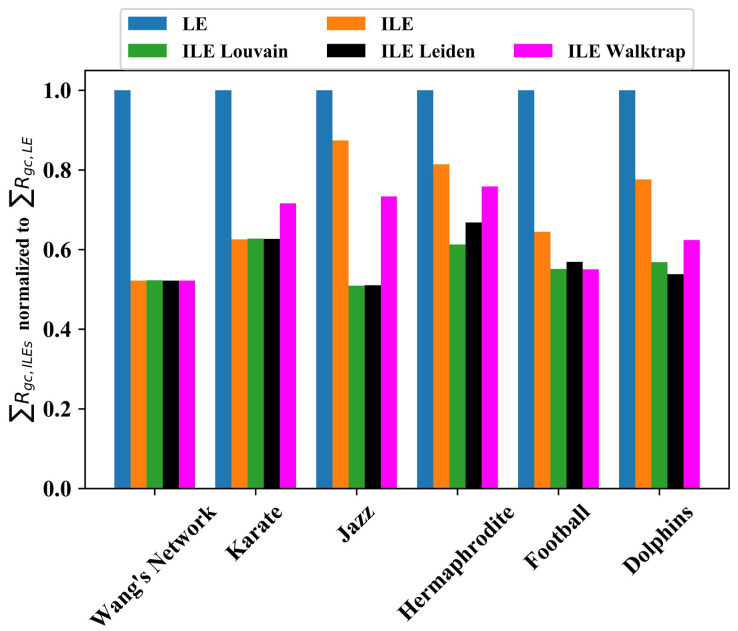
Comparing the performance of edge significance quantification algorithms, LE [46], the proposed ILE with fixed number of communities, and ILE versions with dynamic community number detection in each iteration. The bar plots show the area under the $R_{GC}$ curve found by each algorithm normalized to the result obtained by the LE algorithm on the discussed benchmark networks.

**Table 1 entropy-25-00365-t001:** Comparing the performance of each method by calculating the area under $R_{GC}$ curve as a result of each method on Wang et al.’s network [47].

Method	Area
LE	41.300
ILE	21.566
ILE_Louvain	21.600
ILE_Leiden	21.566
ILE_Walktrap	21.566

**Table 2 entropy-25-00365-t002:** Comparing the performance of each method by calculating the area under the $R_{GC}$ curve as a result of each method on the Zachary’s Karate Club network [55].

Method	Area
LE	44.294
ILE	27.705
ILE_Louvain	27.794
ILE_Leiden	27.764
ILE_Walktrap	31.705

**Table 3 entropy-25-00365-t003:** Comparing the performance of each method by calculating the area under the $R_{GC}$ curve as a result of each method on the Dolphins network [56].

Method	Area
LE	97.016
ILE	75.290
ILE_Louvain	55.145
ILE_Leiden	52.209
ILE_Walktrap	60.548

**Table 4 entropy-25-00365-t004:** Comparing the performance of each method by calculating the area under the $R_{GC}$ curve as a result of each method on the Hermaphrodite network [57].

Method	Area
LE	828.304
ILE	674.345
ILE_Louvain	507.464
ILE_Leiden	553.392
ILE_Walktrap	628.266

**Table 5 entropy-25-00365-t005:** Comparing the performance of each method by calculating the area under the $R_{GC}$ curve as a result of each method on the Jazz network [58].

Method	Area
LE	1845.353
ILE	1613.176
ILE_Louvain	940.005
ILE_Leiden	941.843
ILE_Walktrap	1353.838

## Data Availability

All the results can be reproduced through the presented methods. Network data used in this work are available in the following resources. Wang et al.’s Network Data: https://doi.org/10.1007/s10618-010-0181-y (accessed on 1 September 2022). Zachary’s Karate Club Network Data: https://networkrepository.com/soc-karate.php (accessed on 1 September 2022). Dolphins Network Data: https://networkrepository.com/soc-dolphins.php (accessed on 1 September 2022). Hermaphrodite Network Data: https://networks.skewed.de/net/celegans_2019 (accessed on 1 September 2022). Jazz Network Data: https://networkrepository.com/jazz.php (accessed on 1 September 2022).

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
