# Peer review of "Improved Link Entropy with Dynamic Community Number Detection for Quantifying Significance of Edges in Complex Social Networks"

_entropy, 2023, doi:10.3390/e25020365_

Round 1
Reviewer 1 Report (Previous Reviewer 2)
This manuscript is improved comparing to its previous version. But I think it's still below average. I will not recommend this manuscript to be published in Entropy.
The final judgement is up to the Chief Editor.
Reviewer 2 Report (Previous Reviewer 3)
The authors responded with minimal effort to my comments. For example performance analysis is minimal. In response to my comment, that just point to a case in which the worst algorithm perform twice as slow as the best of the authors' three algorithm. They keep three because none perform better that the others. It is not clear how quickly difference growth with the size of data, what kind of networks perform for this pointed case.
Formally, the paper with edits may pass publication criteria, but not clearly. Too many times, the authors go the easiest way. So I would like to see before publication authors' suggestions regarding algorithms and their performance, by stating for what kind of networks each of their algorithms is best suited. This assessment should be supported by significant size examples.
Reviewer 3 Report (Previous Reviewer 1)
The authors worked hard to improve thekr original submission. I am pleased of recommending it for acceptance.
This manuscript is a resubmission of an earlier submission. The following is a list of the peer review reports and author responses from that submission.
Round 1
Reviewer 1 Report
This paper is about the computation of edge importance in large graphs. The authors start form an existing method called Link Entropy (see Ref. 46 in the paper) and discuss an interesting and powerful generalization. The proposed approach has been then tested in conjunction with popular community detection methods such as Louvain, Leiden and Walktrap. Experiments on some real datasets are proposed.
I liked a lot the paper. It is clear and well written, Reference list is huge with more than 60 items, thus signaling the careful review of the state of the art made by the authors. The paper is clear, well-written and easy to follow. Thus I am pleased of recommending it for acceptance. I've just few minor question that can be considered by authors as future work.:
1) What is the computational complexity of the proposed method?
2) Is it possible to consider larger networks in the experimental validation?
3) Is the proposed approach applicable also to directed graphs?
Reviewer 2 Report
The authors seems to want to quantify edge significance via improved link entropy with dynamic community detection in complex social network. If so, the title is confusing and not accurately describes the main content of this paper.
Quantifying edge significance is an important topic in network science field. But this paper is far from a standardized scientific research.
Figures are not explained in detail.
Figures are all not qualified to be present in a research paper. e.g. Lines are overlap that cannot be distinguished each other.
The total quality of the paper is far below average. Therefore, I will not recommend this paper be published in Entropy.
Reviewer 3 Report
The paper advances state of the art just a bit, but not much because the contribution of authors is adopted existing methods to a change of one the solution steps. The results are not sufficiently presented and therefore inconclusive. The presented results are not convincing due to the lack of execution times for different methods, and small networks used to benchmark the results. Combined with the weak originality of the contribution, and lack of comparison of complexity of the several new solutions, and small differences between different versions of improvements measured on small networks would require a big rewrite, beyond just major revision.